# Development of Novel Tamsulosin Pellet-Loaded Oral Disintegrating Tablet Bioequivalent to Commercial Capsule in Beagle Dogs Using Microcrystalline Cellulose and Mannitol

**DOI:** 10.3390/ijms242015393

**Published:** 2023-10-20

**Authors:** Hyuk Jun Cho, Jung Suk Kim, Sung Giu Jin, Han-Gon Choi

**Affiliations:** 1College of Pharmacy, Hanyang University, 55 Hanyangdaehak-ro, Sangnok-gu, Ansan 15588, Republic of Korea; 2Pharmaceutical Research Centre, Hanmi Pharmaceutical Co., Ltd., Paltan-Myeon, Hwaseong 18536, Republic of Korea; 3Department of Pharmaceutical Engineering, Dankook University, 119 Dandae-ro, Dongnam-gu, Cheonan 31116, Republic of Korea

**Keywords:** tamsulosin hydrochloride, microcrystalline cellulose, mannitol, oral disintegrating tablet, multi-unit pellet system, dissolution, bioequivalence

## Abstract

In this study, we developed a tamsulosin pellet-loaded orally disintegrating tablet (ODT) that is bioequivalent to commercially available products and has improved patient compliance using microcrystalline cellulose (MCC) and mannitol. Utilizing the fluid bed technique, the drug, sustained release (SR) layer, and enteric layer were sequentially prepared by coating MCC pellets with the drug, HPMC, Kollicoat, and a mixture of Eudragit L and Eudragit NE, respectively, resulting in the production of tamsulosin pellets. The tamsulosin pellet, composed of the MCC pellet, drug layer, SR layer, and enteric layer at a weight ratio of 20:0.8:4.95:6.41, was selected because its dissolution was equivalent to that of the commercial capsule. Tamsulosin pellet-loaded ODTs were prepared using tamsulosin pellets and various co-processed excipients. The tamsulosin pellet-loaded ODT composed of tamsulosin pellets, mannitol–MCC mixture, silicon dioxide, and magnesium stearate at a weight ratio of 32.16:161.84:4.0:2.0 gave the best protective effect on the coating process and a dissolution profile similar to that of the commercial capsule. Finally, no significant differences in beagle dogs were observed in pharmacokinetic parameters, suggesting that they were bioequivalent. In conclusion, tamsulosin pellet-loaded ODTs could be a potential alternative to commercial capsules, improving patient compliance.

## 1. Introduction

Prostate-related diseases pose a threat to men’s health, and benign prostatic hyperplasia (BPH), in particular, significantly diminishes the quality of life [1]. BPH is a progressive disease with symptoms including dysuria, reduced urinary flow rate, and increased urinary infection [2]. Tamsulosin (hydrochloride), an α1-adrenoceptor antagonist, is a clinically available drug for the treatment of BPH since it primarily relaxes the smooth muscles of the prostate and bladder [3]. This results in an increased maximum urine flow rate reduced post-void residual urine volume, and mitigation of BPH-related symptom progression [2,3]. This drug was initially developed as an immediate-release formulation for early clinical trials; however, this formulation led to a high maximum plasma concentration (C_max_), causing orthostatic hypotension in healthy subjects. To reduce the C_max_ and side effects, commercial products with sustained release (SR) and enteric release properties, such as Flomax^®^ capsule (tamsulosin 0.4 mg) and Omnic^®^ capsule (tamsulosin 0.4 mg), have been developed [4].

However, elderly patients have severe difficulty swallowing conventional solid dosage forms [5]. In elderly individuals, 40% of patients suffer from BPH and dysphagia, an age-related disease [6]. However, tamsulosin is frequently prescribed to elderly patients. Thus, to solve the problem, a new commercial product containing tamsulosin needs to be developed. For rapid disintegration in the oral cavity, an orally disintegrating tablet (ODT) is a convenient formulation for elderly patients [7,8]. A multiunit pellet system (MUPS) can effectively convert general oral dosage forms, such as capsules and tablets, into ODT [9]. This MUPS has a relatively high production yield compared to capsule filling and is an easy procedure for tablet splitting without burst release concerns [10].

In this study, a tamsulosin-pellet-loaded ODT with bioequivalence to a commercial product was developed to address this problem using microcrystalline cellulose (MCC) and mannitol. Using the fluid bed technique, tamsulosin pellets were prepared by sequentially coating the drug, sustained release (SR), and enteric layers. Moreover, the effect of each subsequent layer on drug dissolution was evaluated and compared with that of a commercial capsule. Furthermore, tamsulosin pellet-loaded ODTs were prepared using the selected tamsulosin pellets, and their dissolution and pharmacokinetics were evaluated in beagle dogs.

## 2. Results and Discussion

### 2.1. Tamsulosin Pellets

In our study, a tamsulosin pellet-loaded ODT was developed to enhance patient compliance using MCC and mannitol. First, the drug layer was manufactured by coating the MCC pellets with the drug, talc, and HPMC. HPMC and talc were employed as coating and anti-sticking agents, respectively [11]. The molecular structure of tamsulosin is depicted in Figure 1. The secondary amine (pK_a_ = 8.37) and sulfonamide groups (pK_a_ = 10.23) serve as proton binding sites [12]. These groups could enhance the solubility of the tamsulosin under acidic conditions compared to a neutral environment. When a drug is orally administered, there are changes in pH as it passes from the stomach to the intestine. This shift affects the drug solubility and, eventually, influences its oral bioavailability. Thus, it is crucial to investigate the release profile of tamsulosin in vitro based on pH.

Various SR layers were prepared by coating this drug layer with various amounts of Kollicoat, and their dissolution was investigated at pH 1.2 and pH 6.8 and compared to that of the drug layer (Figure 2). The amount of subsequent SR layer on the drug layer was 15–24%, indicated as a percentage of drug layer mass [13,14]. The subsequent SR layer exhibited a faster dissolution rate at pH 1.2 compared with the commercial capsule, indicating a pH-dependent dissolution profile (Figure 2A) [15,16]. This suggests the need for an additional enteric coating layer to obtain an equivalent to a commercial capsule; hence, the similarity of its dissolution profile at pH 1.2 should be assessed after enteric coating. Tamsulosin was completely dissolved from the drug layer within 30 min at pH 6.8. However, by constructing the subsequent SR layer, the dissolution of tamsulosin was sustained for 6 h at pH 6.8 (Figure 2B). According to the Guidelines on the Investigation of Bioequivalence, the dissolution profiles of the SR layer at pH 6.8 were compared [17,18]. The difference factor (f_1_) and similarity factor (f_2_) were determined: f_1_ = [∑│R_t_ − T_t_│/∑ R_t_] × 100; f_2_ = 50 × log{[1 + (1/n)∑ (R_t_ − T_t_)2]−0.5 × 100}, where n represents the number of comparison time points and T_t_ and R_t_ are the dissolved percentages at each time for the SR layer and the commercial capsule, respectively [19]. In this study, 0 < f_1_ < 15 and 50 < f_2_ < 100 were regarded as having similar dissolution patterns between the two formulations [19,20]. The f_1_ and f_2_ values between the SR layer and the commercial capsule were 36.4 and 30.6 at 15%, 25.2 and 38.3 at 19%, 14.8 and 47.1 at 21%, and 11.4 and 56.2 at 24%, respectively. Among the SR layers tested, the 24% SR layer provided the most similar dissolution pattern; therefore, this SR layer was chosen.

Continuously, various tamsulosin pellets were prepared by coating this SR layer with 15–35% of a mixture of Eudragit L and Eudragit NE (1:1, volume ratio) as the enteric coating agent. In the coating process, triethyl citrate and talc were used as plasticizers and anti-sticking agents, respectively [21]. In general, Eudragit NE is used to form highly flexible films and complement the brittle character of Eudragit L [22]. Their dissolution was assessed at pH 1.2 compared to the above-selected SR layer (Figure 3). The SR layer had a rapid dissolution rate of over 50% within 2 h. As the thickness of the enteric layer increased, the dissolution rate of the tamsulosin pellets decreased. Unlike the others, the 25% enteric layer (f_1_, 14.9; f_2_, 86.8) had a similar dissolution profile to commercial capsules; therefore, the 25% enteric layer was chosen.

Based on those results, a tamsulosin pellet composed of the MCC pellet, drug layer, SR layer, and enteric layer at a weight ratio of 20:0.8:4.95:6.41 was selected as the optimal formulation for the development of tamsulosin pellet-loaded ODT.

The morphologies of the progressive pellets after each coating step are shown in Figure 4. All pellets exhibited irregular spheroid shapes and bumpy surfaces, although some aggregates were observed [23]. The particle size of the pellets gradually increased with the addition of each coating layer [24,25]. In the enteric layer of the final tamsulosin pellet, the portion of aggregates that were separated using a sieve with 250 µm pore size was less than 5%. Additionally, the tamsulosin pellets were freely passed through mesh with a 300 µm pore size. The recommended particle sizes of general pellets were reported to be approximately less than 350 µm due to their smooth feeling in the mouth [26].

The physical state of the tamsulosin in the pellet was analyzed using differential scanning calorimetry (DSC) and powder X-ray diffraction (PXRD), as shown in Figure 5a,b. In the DSC thermograms, the tamsulosin exhibited a sharp peak at its melting point (230 °C), whereas the blank pellet showed no such peak. The corresponding peak was observed in the tamsulosin pellet. According to the PXRD results, the drug displayed its unique patterns over the scanned range. These patterns were not present in the blank pellet; however, they were detected in the tamsulosin pellet. Overall, the DSC and PXRD results indicated that tamsulosin is crystalline and maintained its crystallinity during the preparation of the tamsulosin pellet. Moreover, Fourier transform infrared spectra (FT-IR) were obtained, and the results are shown in Figure 5c. Tamsulosin showed distinctive peaks at approximately 1350 cm^−1^ (S=O stretching), 1250 cm^−1^ (C-O stretching), and 1640 cm^−1^ (C=C stretching). Contrary to the blank pellet, these signals were shown in the tamsulosin pellet, confirming the presence of tamsulosin in the pellet.

### 2.2. Tamsulosin Pellet-Loaded ODTs

To choose an appropriate co-processed excipient, the tamsulosin pellet, various co-processed excipients, silicon dioxide, and magnesium stearate were completely blended and compressed, leading to the production of tamsulosin pellet-loaded ODTs (Table 1; compositions I–III), and their dissolution tests were performed at pH 1.2 compared to the commercial capsules (Figure 6). There are possible risks of cracking on the coated layer during the compression process, leading to the accelerated dissolution of ODTs [27]. To prevent this, the mannitol·MCC mixture, mannitol·croscarmellose sodium mixture, and mannitol·povidone mixture were employed as the co-processed excipients. Magnesium stearate and silicon dioxide were used as the lubricants. The protective effect of excipients on ODTs was evaluated by their dissolution profile (Figure 6A); retarded dissolution may indicate an excellent protective effect [28]. All compositions increased the dissolution rates of the drug compared to the tamsulosin pellets. These results suggest that the compression process on tamsulosin pellets resulted in the cracking of the coated layer and increased the dissolution of ODTs. Moreover, composition I, prepared with the mannitol–MCC mixture, showed lower dissolution rates than compositions II and III, prepared with the others. Therefore, the mannitol–MCC mixture was selected because of its protective effects. To determine the amounts of excipients (mannitol–MCC mixture and silicon dioxide), ODTs were prepared with various amounts of the mannitol–MCC mixture (Table 1; compositions I, IV, V, and VI), and their dissolution was evaluated (Figure 6B). Similarly, they increased the dissolution rate of the drug compared to that of the tamsulosin pellet. Moreover, compositions V and VI had significantly lower dissolution rates than compositions I and IV; however, the dissolution rates of compositions V and VI were not significantly different. Thus, the composition V, composed of tamsulosin pellet, mannitol·MCC mixture, silicon dioxide, and magnesium stearate at a weight ratio of 32.16:161.84:4.0:2.0 was chosen as an appropriate formulation of the tamsulosin pellet-loaded ODT. Mannitol has frequently been employed in the development of ODT formulations owing to its excellent water solubility, good mouthfeel, taste, and fast disintegration [29]. However, it induced brittle deformation behavior. MCC with highly plastic characteristics markedly protects the cracking of the coating layer against compaction impacts [27,30]. Moreover, silicon dioxide, a lubricant, can undergo plastic deformation and protect pellets by enhancing flowability and compressibility [31,32]. In our study, MCC and silicon dioxide in ODTs increased their protective effects [33]. Furthermore, the flow properties of the composition V powder were assessed by measuring bulk density, tapped density, Hausner ratio, and angle of repose. The bulk density (0.59 ± 0.02 g/mL), tapped density (0.69 ± 0.1 g/mL), Hausner ratio (1.17 ± 0.06), and angle of repose (32.0 ± 2.1°) supported its good flowability.

As shown in Figure 7, the dissolution profiles of the tamsulosin pellet-loaded ODT and the commercial capsule were compared at pH 1.2 and 6.8, and their dissolution similarity was evaluated, as mentioned above. Their f_1_ and f_2_ values were 2.3 and 98.6 at pH 1.2 and 5.6 and 67.9 at pH 6.8, respectively. Hence, our tamsulosin pellet-loaded ODT showed dissolution profiles equivalent to those of the commercial capsules.

Next, the effect of hardness on the dissolution of the drug in the tamsulosin pellet-loaded ODTs was determined. However, in 3–6 KP, the dissolution was not significantly different. Therefore, the compression force on the ODTs only negligibly affected the drug dissolution. Additionally, the disintegration time and friability of tamsulosin pellet-loaded ODTs were assessed. In 3–6 KP, all ODTs were disintegrated within 30 s, and friability was below 0.05%, indicating suitable properties for ODTs. The content uniformity of the tamsulosin pellet-loaded ODT showed 5.80 ± 0.10%, meeting the acceptance criteria of USP <905> Uniformity of dosage units [34].

The stability of the tamsulosin pellet-loaded ODT and the commercial capsule was evaluated by investigating the impurities of tamsulosin under the determined conditions (temperature: 40 °C; relative humidity: 75%) for 6 months. The total impurities in pellet-loaded ODT were lower than those in the commercial capsule (initial, 0.04 ± 0.01% vs. 0.07 ± 0.01%; 3 months, 0.21 ± 0.03% vs. 0.34 ± 0.05%; 6 months, 0.49 ± 0.01% vs. 0.65 ± 0.01%, respectively.) Therefore, tamsulosin pellet-loaded ODT was more stable than the commercial capsule.

### 2.3. Pharmacokinetics

The mean plasma concentration-time profiles of tamsulosin after the oral administration of the pellet-loaded ODT and commercial capsules at an equivalent dose of 0.4 mg tamsulosin in beagle dogs are illustrated in Figure 8. The tamsulosin pellet-loaded ODT and commercial capsules showed no significant difference in plasma concentration at each time point [34,35]. The corresponding pharmacokinetic parameters are listed in Table 2. The Tmax of a commercial tamsulosin-loaded immediate-release formulation in beagle dogs has been reported to be within 1 h [36]. In this study, both formulations showed an increased T_max_, indicating a sustained release pattern [37]. Moreover, no significant differences were observed in their pharmacokinetic parameters, including AUC, C_max_, and T_max_ (45.988 ± 15.345 vs. 46.251 ± 14.245 h·ng/mL; 7.237 ± 2.324 vs. 7.625 ± 2.634 ng/mL; 2.833 ± 0.718 vs. 2.917 ± 0.515 h, respectively). Therefore, our tamsulosin pellet-loaded ODT was bioequivalent to a commercial capsule in beagle dogs [38].

## 3. Materials and Methods

### 3.1. Materials

Tamsulosin hydrochloride (abbreviated as “tamsulosin”) was kindly acquired from Hanmi Pharm. Co. (Hwasung, Republic of Korea). Microcrystalline cellulose pellet (Cellet 175^®^; abbreviated as “MCC pellet”) was purchased from IPC Process-Center GmbH (Dresden, Germany). Hydroxypropylmethylcellulose (HPMC 2910) was procured from Lotte Fine Chemical Co. (Incheon, Republic of Korea). Triethyl citrate, talc, silicon dioxide, and magnesium stearate were purchased from Hanmi Pharmaceutical. Co. (Hwasung, Republic of Korea). Kollicoat SR 30D (polyvinyl acetate 30% aqueous dispersion; abbreviated as “Kollicoat”) was supplied by BASF (Ludwigshafen, Germany). Eudragit NE 30D (acrylate and methyl methacrylate copolymer 30% aqueous dispersion; abbreviated as “Eudragit NE”) and Eudragit L 30D-55 (methacrylic acid and ethyl acrylate copolymer 30% aqueous dispersion; abbreviated as “Eudragit L”) were purchased from Evonik Nutrition & Care GmbH (Essen, Germany). Mannitol·MCC mixture (F-melt type C), mannitol·croscarmellose sodium mixture (Parteck ODT), and mannitol·povidone mixture (Orocell) were obtained from Fuji Chemical Co. (Osaka, Japan), Merck KGaA (Darmstadt, Germany), and Pharmatrans-Sanaq AG (Basel, Switzerland), respectively. The commercial capsule (Hanmi Tams^®^) containing tamsulosin 0.4 mg, a reference-listed drug in Korea, was supplied by Hanmi Pharm. Co. (Hwaseong, Republic of Korea). All other chemicals were of reagent grade and were used without further purification.

### 3.2. Preparation of Tamsulosin Pellets

#### 3.2.1. Drug Layer

The coating solution was prepared by dissolving/dispersing 3 g tamsulosin, 1.5 g HPMC, and 1.5 g talc in 270 mL of 90% ethanol. The MCC pellet (particle size of 150–200 µm) (150 g), an inert core, was coated with a coating solution using a fluid bed coater (GRETC-30; GR-ENG, Hwaseong, Republic of Korea). The fluid bed coater was assembled in bottom-spray mode and coated with the inlet air temperature of 32–38 °C and product temperature of 27–33 °C. The spray rate was set to 2.7 g/min with 1.0 bar of atomizing air pressure. After spraying, the coated pellets were subjected to continuous fluidization and dried for 10 min.

#### 3.2.2. Subsequent SR Layer

The SR layer was subsequently deposited on the pellets by coating them with a dispersed solution composed of Kollicoat (105 g), talc (3.15 g), and distilled water (110 mL). The coating solution was mixed using a magnetic stirrer at 200 rpm for 1 h, and agitation was followed during the fluid bed coating process. The fluid bed coater was operated with inlet air and product temperatures of 38–42 °C and 30–34 °C, respectively. The spray rate and atomizing air pressure are maintained under the conditions described in Section 3.2.1. The subsequent SR layer was fluidized and dried at an inlet temperature of 50 °C for 1 h.

#### 3.2.3. Tamsulosin Pellets

Similarly, pellets prepared with the subsequent SR layer were coated with the enteric coating solution using the fluid-bed technique. The enteric coating solution was prepared by mixing Eudragit L (56.7 g), Eudragit NE (5.67 g), triethyl citrate (3.4 g), and talc (7.5 g) in distilled water (113 mL). Except for the spray rate, the same process parameters were applied as described in Section 3.2.1. The enteric coating solution was sprayed at a speed of 1.6 g/min. The enteric layer was fluidized and dried at an inlet temperature of 50 °C for 30 min, manufacturing the tamsulosin pellets.

### 3.3. Preparation of Tamsulosin Pellet-Loaded ODTs

The compositions of the ODTs are presented in Table 1. The tamsulosin pellets were mixed with various co-processed excipients and silicon dioxide for 20 min at 16 rpm using a blender (Bin Type; Y-tech, Hwasung, Republic of Korea). Subsequently, magnesium stearate was blended in the blender for 5 min at 16 rpm. The resulting mixture was compressed into round tablets with a diameter of 9 mm using a tableting machine (single-punch type) (Autotab 200TR; Ichihashi Seiki, Kyoto, Japan).

### 3.4. Flow Properties

The bulk volume (V_0_) of the composition V powder (weighing 10 g) was assessed using a 50 mL measuring cylinder. After 1500 taps, the tapped volume (V_1500_) was measured. Both bulk and tapped densities were calculated as 10 g/V_0_ and 10 g/V_1500_, respectively. The Hausner ratio was obtained by dividing the tapped density by the bulk density.

The angle of repose was determined using the fixed-height funnel method. After pouring 5 g of composition V powder, a conical pile was formed. The height and radius of the pile were then measured, and the angle of repose was calculated using the inverse tangent value.

### 3.5. Morphology, Differential Scanning Calorimetry (DSC), Powder X-ray Diffraction (PXRD), Fourier Transform Infrared Spectroscopy (FT-IR)

The morphologies of the drug layer, subsequent SR layer, and final tamsulosin pellet were evaluated using scanning electron microscopy (SEM). Each pellet was fixed using conductive double-sided carbon tape on a brass specimen stub, and SEM analysis was conducted using a tabletop microscope (TM3000; Hitachi, Fukuoka, Japan) operating at a voltage of 15 kV.

DSC was conducted using a Q20 (TA instrument, New Castle, DE, USA) for thermal characterization. Approximately 5 mg of each powder, including tamsulosin, blank pellet, and tamsulosin pellet, was placed in an aluminum crucible and sealed with a hermetic cover. The analysis temperature was increased from 120 to 300 °C at a heating rate of 10 °C/min under a nitrogen purge flow of 20 mL/min.

PXRD was performed using a RINT2000 (Rigaku Corporation, Tokyo, Japan) to obtain the crystallinity of tamsulosin. PXRD analysis was carried out using monochromatic Cu Kα radiation (λ = 1.5406 Å) at 100 mA and 40 kV. Each drug was scanned from 3.0° to 50° (2θ value) with an increment of 0.02°/s.

FT-IR was analyzed to confirm the presence of tamsulosin in the pellet by using a spectrometer (Frontier; PerkinElmer, Waltham, MA, USA). The samples were scanned from 400 to 2000 cm^−1^ at a resolution of 4 cm^−1^.

### 3.6. Dissolution

The dissolution of tamsulosin was performed using a USP dissolution apparatus II (PTWS-1220; Pharma Test Apparatus AG, Darmstadt, Germany). Briefly, 900 mL of pH 1.2 and pH 6.8 were used as the dissolution media. The temperature was adjusted to 37 ± 0.5 °C. Tamsulosin pellet-loaded ODT and commercial capsules, the equivalent of 0.4 mg of tamsulosin, were placed in the dissolution medium at a paddle speed of 75 rpm. At a predetermined time, 2 mL of the medium was sampled and filtered through a PVDC membrane filter (0.45 µm). High-performance liquid chromatography (HPLC) (Hitachi, Tokyo, Japan) was used to analyze the amount of tamsulosin. A mixture of acetonitrile and pH 2.0 buffer solution (2:5, *v*/*v*) was used as the mobile phase. To prepare a buffer solution (pH 2.0), 6.2 mL of perchloric acid and 1.5 g of sodium hydroxide were dissolved in 1 L of distilled water, and the pH was 2.0 with sodium hydroxide. Inertsil^®^ ODS-2 column (150 mm × 4.6 mm, 5 μm) was placed in a column oven. The temperature of the column oven was maintained at 40 °C, and the sample solution (400 μL) was injected. The flow rate of the mobile phase was 1.0 mL/min, and the absorbance was monitored at 225 nm. A calibration curve covering a range of 0.004–1.00 μg/mL provided good linearity with an R^2^ value > 0.999. Moreover, the accuracy and precision at three different concentrations were less than 2.0%.

### 3.7. Uniformity of Content

The content uniformity of the tamsulosin pellet-loaded ODTs was assessed following USP <905> Uniformity of dosage units [34]. Ten ODTs were randomly selected, and each ODT was dissolved in 50 mL of a mixture of methanol, acetonitrile, and pH 2.0 buffer solution (3:2:5, *v*/*v*) and vortexed for 30 min. The solution was diluted tenfold and filtered through a 0.45 µm PVDC membrane filter. The amount of tamsulosin in each ODT was determined by HPLC, as described in Section 3.5.

### 3.8. Disintegration and Friability

The disintegration test was performed using a USP disintegration tester (DIT-200; Labfine INC, Gunpo, Republic of Korea). Distilled water (37.0 ± 0.5 °C) was used as a disintegration medium, and the disintegration time was measured when the lump disappeared.

The friability was assessed by the weight changes in ODTs. The ODTs were placed in a friability tester (TAR 200; Erweka GmbH, Darmstadt, Germany) and rotated at a speed of 25 rpm for 4 min. Next, the weight of the remaining ODTs was accurately weighted, excluding any separated particles.

### 3.9. Stability

Tamsulosin pellet-loaded ODT, and the commercial capsule were packed into aluminum-aluminum blisters using a packaging machine (Fantasy Plus; O.M.A.R. S.r.l., Milan, Italy). They were stored for 6 months at a temperature of 40 °C and a relative humidity of 75%. The impurities in tamsulosin were evaluated by analyzing its degradation. Each tablet was added to 50 mL of a mixture comprising methanol, acetonitrile, and pH 2.0 buffer solution (20:24:56, *v*/*v*) and vortexed for 30 min. The solution underwent filtration through a 0.45 µm PVDC membrane filter. The impurities of tamsulosin were determined by HPLC, as described method in Section 3.5, though the mobile phase was different. A mixture of acetonitrile and pH 2.0 buffer solution (3:7, *v*/*v*) was used as the mobile phase. To prepare a buffer solution (pH 2.0), 4.3 mL of perchloric acid and 1.5 g of sodium hydroxide were dissolved in 1 L of distilled water. Subsequently, it was adjusted to pH 2.0 using sodium hydroxide. A calibration curve covering a range of 0.008–0.13 μg/mL provided good linearity with an R^2^ value > 0.999.

### 3.10. Pharmacokinetics

#### 3.10.1. Administration in Beagle Dogs

In accordance with the NIH Policy and Animal Welfare Act, pharmacokinetic tests were performed following approval from the Institutional Animal Care and Use Committee (IACUC) at the Hanmi Research Center (AEC-20080430-0006). Twelve male beagles (8–12 kg) were divided into two groups. During the 24 h experimental period, each dog was individually placed in a cage and fasted overnight. In each group, the dogs were orally administered tamsulosin pellet-loaded ODT and a commercial capsule, both equivalent to 0.4 mg of tamsulosin. Blood samples (approximately 1.5 mL) were collected from the cephalic veins. These samples were immediately centrifuged at 12,000× *g* for 3 min at 4 °C using a centrifuge (5415C; Eppendorf, Hauppauge, NY, USA). The plasma was separated and stored at −80 °C until further analysis.

#### 3.10.2. Plasma Sample Preparation and Analysis

The plasma samples (200 μL) were mixed with 20 μL of doxazosin solution (200 ng/mL) as an internal standard. Methyl tert-butyl ether (1 mL) was then added and mixed for 10 min using a vortex mixer. To separate the supernatant, the mixture was centrifuged at 12,000× *g* for 3 min using a centrifuge. Subsequently, the solvent was removed under a nitrogen atmosphere using an evaporator (MG-2200; Eyela, Tokyo, Japan). For reconstitution, 150 μL of acetonitrile and 10 mM of ammonium acetate (80:20) containing 0.1% formic acid were used. The supernatant of the reconstituted solution was separated by centrifuging at 12,000× *g* for 3 min, and 10 μL of each sample was analyzed by liquid chromatography–tandem mass spectrometry (XEVO TQ-S; Waters, Massachusetts, MA, USA). The flow rate of the mobile phase, which was the same as that of the reconstituted solution, was 0.18 mL/min. The temperature of the column oven was kept at 35 °C, and the reverse phase C18 Column (HSS T3, 100 mm × 2.1 mm, 1.7 μm) was assembled. The capillary voltage and collision energy were 3.5 kV and 20 V, respectively. The cone voltages for tamsulosin and doxazosin were 20 V and 30 V, respectively. The desolvation and cone gas flow rates were 850 and 50 L/h, respectively. The desolvation and source temperatures were 400 °C and 150 °C, respectively. A calibration curve covering a range of 0.1–50 ng/mL showed good linearity with an R2 value > 0.999.

## 4. Conclusions

The tamsulosin pellet-loaded ODT composed of tamsulosin pellet, mannitol·MCC mixture, silicon dioxide, and magnesium stearate at a weight ratio of 32.16:161.84:4.0:2.0 provided the best protective effect in terms of the coating and similar dissolution profile compared to the commercial capsule. Moreover, no significant differences in beagle dogs were observed in the pharmacokinetic parameters, including AUC, C_max_, and T_max_, suggesting that they were bioequivalent. In conclusion, tamsulosin-loaded ODTs using MCC and mannitol are a potential alternative to commercial pellets for improving patient compliance.

## Figures and Tables

**Figure 1 ijms-24-15393-f001:**
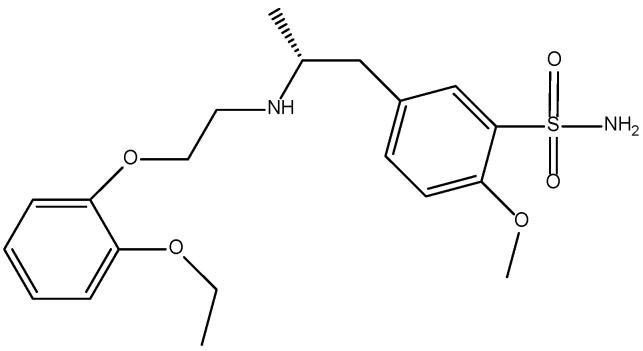
Molecular structure of tamsulosin.

**Figure 2 ijms-24-15393-f002:**
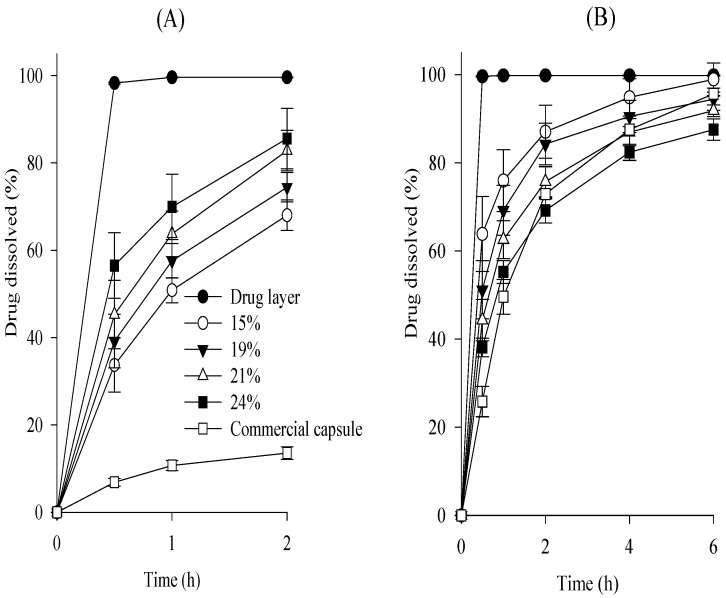
Effect of coating amounts on the dissolution of drug in the subsequent SR layer in pH 1.2 (**A**) and pH 6.8 (**B**). Each value represents the mean ± S.D. (n = 6).

**Figure 3 ijms-24-15393-f003:**
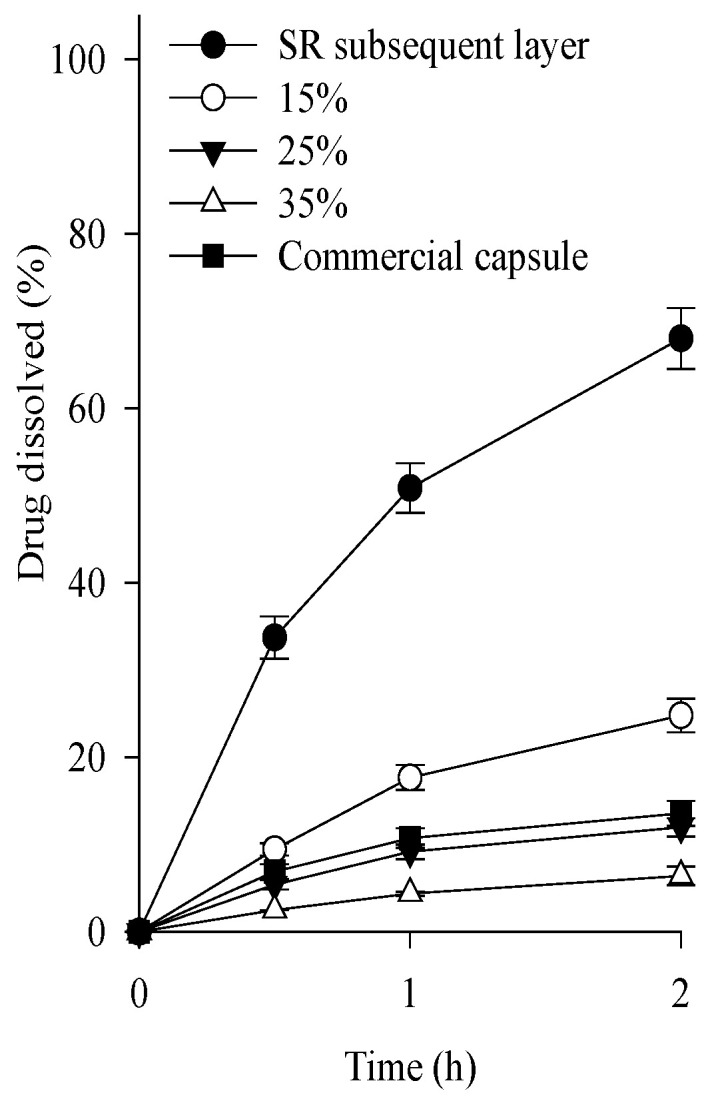
Effect of coating amounts on the dissolution of drug in the tamsulosin pellet in pH 1.2. Each value represents the mean ± S.D. (n = 6).

**Figure 4 ijms-24-15393-f004:**

Scanning electron microscopy (SEM) images (×50): (**a**) drug layer; (**b**) subsequent SR layer; (**c**) tamsulosin pellet.

**Figure 5 ijms-24-15393-f005:**
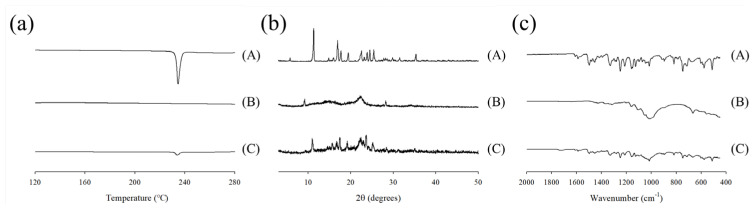
Differential scanning calorimetry (DSC) thermograms (**a**), powder X–ray diffraction (PXRD) patterns (**b**), and Fourier transform infrared spectroscopy (FT–IR) spectra (**c**); (A) tamsulosin; (B) blank pellet; (C) tamsulosin pellet.

**Figure 6 ijms-24-15393-f006:**
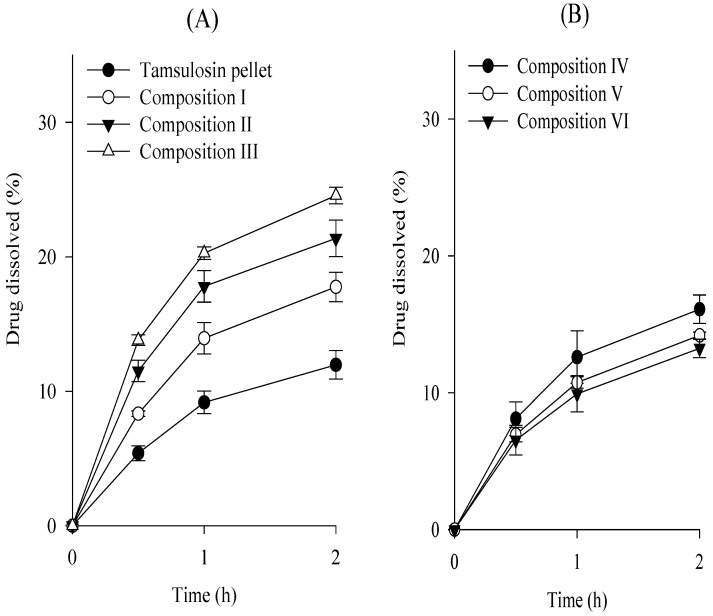
Effect of excipients (**A**) and excipient amounts (**B**) on the dissolution of drug in the tamsulosin pellet-loaded ODTs in pH 1.2. Each value represents the mean ± S.D. (n = 6).

**Figure 7 ijms-24-15393-f007:**
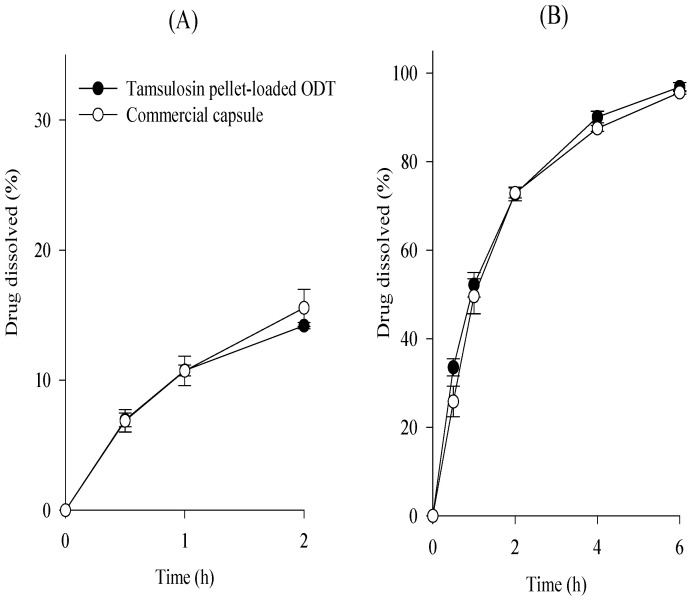
Comparative dissolution profiles of tamsulosin pellet-loaded ODTs and commercial capsule in pH 1.2 (**A**) and pH 6.8 (**B**). Each value represents the mean ± S.D. (n = 6).

**Figure 8 ijms-24-15393-f008:**
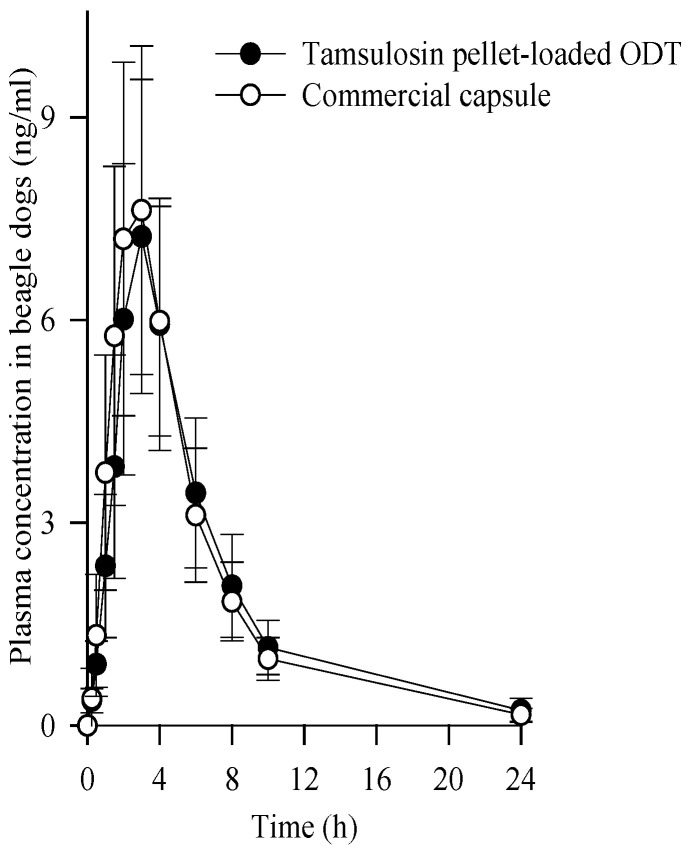
Plasma concentration-time profiles of tamsulosin after the oral administration of tamsulosin pellet-loaded ODT and commercial capsule at the equivalent dose of 0.4 mg tamsulosin in beagle dogs. Each value represents the mean ± S.D. (n = 6).

**Table 1 ijms-24-15393-t001:** Composition of tamsulosin pellet-loaded ODTs. * Equivalent to 0.4 mg tamsulosin.

Ingredients	I	II	III	IV	V	VI
Tamsulosin pellet	32.16 *	32.16	32.16	32.16	32.16	32.16
Mannitol·MCC mixture	165.84	-	-	163.84	161.84	159.84
Mannitol·croscarmellose sodium mixture	-	165.84	-	-	-	-
Mannitol·povidone mixture	-	-	165.84	-	-	-
Silicon dioxide	-	-	-	2.0	4.0	6.0
Magnesium stearate	2.0	2.0	2.0	2.0	2.0	2.0
Total weight (mg)	200.0	200.0	200.0	200.0	200.0	200.0

**Table 2 ijms-24-15393-t002:** Pharmacokinetic parameters after the oral administration of tamsulosin pellet-loaded ODT and commercial capsule in beagle dogs. Each value represents the mean ± S.D. (n = 6). The tamsulosin pellet-loaded ODT and commercial capsule contained 0.4 mg tamsulosin.

Parameter	Tamsulosin Pellet-Loaded ODT	Commercial Capsule
T_max_ (h)	2.833 ± 0.718	2.917 ± 0.515
C_max_ (ng/mL)	7.237 ± 2.324	7.625 ± 2.634
AUC (h·ng/mL)	45.988 ± 15.345	46.251 ± 14.245
t_1/2_ (h)	4.311 ± 0.537	3.832 ± 0.520
K_el_ (h^−1^)	0.161 ± 0.020	0.181 ± 0.025

## Data Availability

Data available on request due to restrictions e.g., privacy or ethical.

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
