# Peer review of "Development of Novel Tamsulosin Pellet-Loaded Oral Disintegrating Tablet Bioequivalent to Commercial Capsule in Beagle Dogs Using Microcrystalline Cellulose and Mannitol"

_ijms, 2023, doi:10.3390/ijms242015393_

Round 1
Reviewer 1 Report
The author should provide another characterization parameter for the pellet and ODT such as morphology, sphericity, micrometric properties, friability, entrapment efficiency, uniformity of content, and in vitro release studies.
The author should provide IR for the pure drug, blank pellet, and drug-loaded pellet to confirm the presence of drug.
The author should provide DSC or XRD to identity the physical state of formulation.
The author should provide all the evaluation parameters for the tablets.
Stability studies should be carried out.
Author Response
Point 1. The author should provide another characterization parameter for the pellet and ODT such as morphology, sphericity, micrometric properties, friability, entrapment efficiency, uniformity of content, and in vitro release studies.
Response 1:
- Added “Furthermore, the --- flowability.”, “Additionally, the ---dosage units [34].”, “3.4. Flow properties”, “3.7. Uniformity of content” and “3.8. Disintegration and friability” in line 177-180, line 196-200, line 283-291, line 329-335 and line 336-343, respectively.
Point 2. The author should provide IR for the pure drug, blank pellet, and drug-loaded pellet to confirm the presence of drug.
Response 2:
- Added “Moreover, fourier --- the pellet” in line 133-137 and provided FT-IR results in Fig. 5-(c).
Point 3. The author should provide DSC or XRD to identity the physical state of formulation.
Response 3:
- Added “The physical --- pellet” in line 125-133 and provided DSC and XRD data in Fig. 5-(a) and (b).
Point 4. Stability studies should be carried out.
Response 4:
- Added “The stability--- capsule” and “3.9. Stability” in line 201-207 and line 344-357, respectively.

Reviewer 2 Report
Dear authors, This research work entitled "Development of novel tamsulosin pellet-loaded oral disintegrating tablet bioequivalent to commercial capsule in beagle dogs using microcrystalline cellulose and mannitol" is highly impressive and will provide a new horizon towards the treatment of deadly prostate hyperplasia. The sequence of paper, methodology and results are well executed. Congratulations for the good paper.
However, I have few suggestions to further enhance the quality of the manuscript
1. Please add some information regarding prostate hyperplasia in the introduction section. Kindly explain the role of your formulation in the mitigation of the severity of this disease.
2. Please add the relevant information and citation in the introduction section from this review article (Nanotreatment and Nanodiagnosis of Prostate Cancer: Recent Updates)
3. Did authors performed any toxicity studies to ascertain the biocompatibility of formulation?
4. FTIR of the formulations will be encouraged
5. SEM images seems to be perfect, showing good PDI as well as zeta potential. Therefore, it is recommended to mention the zeta and PDI findings to accompany the results.
Author Response
Dear authors, This research work entitled "Development of novel tamsulosin pellet-loaded oral disintegrating tablet bioequivalent to commercial capsule in beagle dogs using microcrystalline cellulose and mannitol" is highly impressive and will provide a new horizon towards the treatment of deadly prostate hyperplasia. The sequence of paper, methodology and results are well executed. Congratulations for the good paper.
However, I have few suggestions to further enhance the quality of the manuscript
Point 1. Please add some information regarding prostate hyperplasia in the introduction section. Kindly explain the role of your formulation in the mitigation of the severity of this disease.
Response 1:
- Revised “BPH --- [2, 3].” in line 34-40.
Point 2: Please add the relevant information and citation in the introduction section from this review article (Nanotreatment and Nanodiagnosis of Prostate Cancer: Recent Updates)
Response 2:
- Added “Prostate --- life [1].” in line 33-34.
- Added “Barani et al., 2020” in the section “References”.
Point 3: Did authors performed any toxicity studies to ascertain the biocompatibility of formulation?
Response 3: Thank you for your comments. The authors truly believe the toxicity studies are important. In this study, no novel excipients were used. All excipients are listed in ‘FDA inactive ingredients in approved drug products’ and have been widely used in marked products. Therefore, toxicity study for biocompatibility of formulation was not conducted.
Point 4. FTIR of the formulations will be encouraged
Response 4:
- Added “Moreover, fourier --- the pellet” in line 133-137 and provided FT-IR results in Fig. 5-(c).
Point 5. SEM images seems to be perfect, showing good PDI as well as zeta potential. Therefore, it is recommended to mention the zeta and PDI findings to accompany the results.
Response 5: We really appreciate your comments. Authors believe PDI and zeta potential would support the SEM results. However, the particle sizes of the prepared pellets are over 1 μm. We are really sorry that we could not conduct the experiment due to lack of facilities for measurement of PDI and zeta potential of pellets.

Reviewer 3 Report
The manuscript entitled “ Development of novel tamsulosin pellet-loaded oral disintegrating tablet bioequivalent to commercial capsule in beagle dogs using microcrystalline cellulose and mannitol” is well organized. The manuscript may be accepted for publication contingent on addressing the following comments.
- This research aimed to develop oral disintegrated tablets. The study design focuses on performing drug release profiles in the stomach and intestine pHs. There is no disintegration test was performed for the tablets. A disintegration test needs to be added to support if the authors have successfully developed disintegrated tablets or not.
- In line 256, please clarify the volume of the dissolution medium. Is it 900 liter or 900 mL?
- In line 265, please check the value of the injection volume. A 400 uL is too much as the optimal injection volume for an HPLC column (150 mm × 4.6 mm, 5 µm) is less than 40 uL.
- In Figure 3, 5, and 6, some formulations showed a low percentage release especially in the initial time points. The quantity of tamsulosin in these samples were out of the calibration curve range. Please clarify the analytical method of the in vitro drug release samples.
Author Response
The manuscript entitled “Development of novel tamsulosin pellet-loaded oral disintegrating tablet bioequivalent to commercial capsule in beagle dogs using microcrystalline cellulose and mannitol” is well organized. The manuscript may be accepted for publication contingent on addressing the following comments.
In line 265, please check the value of the injection volume. A 400 uL is too much as the optimal injection volume for an HPLC column (150 mm × 4.6 mm, 5 µm) is less than 40 uL.
In Figure 3, 5, and 6, some formulations showed a low percentage release especially in the initial time points. The quantity of tamsulosin in these samples were out of the calibration curve range. Please clarify the analytical method of the in vitro drug release samples.
Response: Thank you for your review. We have provided responses in “Response 3”.
Point 1. This research aimed to develop oral disintegrated tablets. The study design focuses on performing drug release profiles in the stomach and intestine pHs. There is no disintegration test was performed for the tablets. A disintegration test needs to be added to support if the authors have successfully developed disintegrated tablets or not.
Response 1:
- Added “Additionally, the ---dosage units [34].” and “3.8. Disintegration and friability” in line 196-200. and line 336-343, respectively.
Point 2: In line 256, please clarify the volume of the dissolution medium. Is it 900 liter or 900 mL?
Response 2:
- Revised “900 mL” in line 313.
Point 3: In Figure 3, 5, and 6, some formulations showed a low percentage release especially in the initial time points. The quantity of tamsulosin in these samples were out of the calibration curve range. Please clarify the analytical method of the in vitro drug release samples.
Response 3: Revised “A calibration --- value > 0.999” in line 325-327. We appreciate your comments. The authors believe the injection volume might be considered somehow excessive. However, the amount of tamsulosin was small (0.4mg). Accordingly, the injection volume of 400 μL was used to complement and obtain broader range of calibration curve. We confirmed that the injection volume did not have adversely effects on the column throughout the experiments.

Round 2
Reviewer 1 Report
All comments have been addressed.